# Building a Predictive Model of Social-Emotional Adjustment: Exploring the Relationship between Parenting Self-Efficacy, Parenting Behaviour and Psychological Distress in Mothers of Young Children in Ireland

**DOI:** 10.3390/ijerph18062861

**Published:** 2021-03-11

**Authors:** Sabrina Coyle, Kiran M. Sarma, Catherine Maguire, Leora De Flumere

**Affiliations:** 1Perinatal Mental Health, The Coombe Women and Infants University Hospital, D08 XW7X Dublin, Ireland; 2School of Psychology, National University of Ireland, H91 TK33 Galway, Ireland; kiran.sarma@nuigalway.ie; 3School of Applied Psychology, University College Cork, T12 K8AF Cork, Ireland; Catherine.maguire@ucc.ie; 4School of Psychology, Trinity College, D02 PN40 Dublin, Ireland; leora.deflumere@gmail.com

**Keywords:** social-emotional, psychological distress, parenting, young children, maternal

## Abstract

The purpose of this study was to generate greater understanding of social-emotional difficulties in infants and toddlers in an Irish context. This study compared rates of reported social-emotional difficulties in young children in clinical and non-clinical samples and probed a predictive model of social-emotional adjustment. Data were collected from a cross-sectional sample of 72 mothers of young children aged between 12 and 48 months. Mothers were recruited from waiting lists for child Early Intervention services (clinical sample) and community mother-toddler groups (non-clinical sample). Mothers completed a questionnaire battery which assessed parenting self-efficacy, parenting behaviour, psychological distress and child social-emotional adjustment. The results indicated that 55.5% of young children in the clinical sample and 15% in the non-clinical sample had significant social-emotional problems. Similarly, 55.5% of young children in the clinical sample and 30% in the non-clinical sample had significant delays in the acquisition of social-emotional competencies. Two hierarchical multiple regressions were carried out with social-emotional problems and social-emotional competencies as the respective criterion variables. Clinical or non-clinical group membership, parenting satisfaction and maternal psychological distress were found to be significant predictors of child social-emotional problems in a model which explained 59% of the variance. Task-specific self-efficacy was the only significant predictor of child social-emotional competencies in a model which explained 21% of the variance. The significant rates of social-emotional problems in young children in the current study and the potential negative impact on child health and wellbeing, suggest that the early assessment of social-emotional adjustment should be incorporated into routine clinical assessment for young children. For services to effectively meet the needs of children with social-emotional difficulties and their families, consideration of maternal factors is also necessary.

## 1. Introduction

The developmental tasks of infancy and toddlerhood include the development of self-regulation and the beginning of peer relationships. Research has demonstrated that self-regulatory capacities in infants and toddlers are significantly related to the development of social competence. According to the research, early emerging individual differences in social competence are relatively stable [1,2] and are linked to subsequent adjustment in later childhood [3]. While limited data are available, international literature [4,5,6,7,8,9,10,11] to date has demonstrated that there is a high prevalence of social-emotional difficulties in infants and toddlers with prevalence estimates for 2 and 3 year old children ranging between 7% and 24% with the median falling near 10%. Early deficits in social-emotional functioning have been identified as a significant risk factor for persistent difficulties in children and may have a long-term impact on child developmental outcomes such as social functioning, behavioural competence and academic achievement [12,13,14,15]. Therefore, it is necessary to examine infant and toddler social emotional development in order to provide timely, effective intervention.

Reasons for the dearth of research examining prevalence rates of social-emotional difficulties, particularly in young children, may be numerous. One potential reason concerns the difficulty in recognizing and detecting internalizing type difficulties in infants and toddlers, resulting in an under-recognition of social-emotional problems [16]. Externalizing difficulties are more reliably observed as they are aversive, overt and interpersonal. Another reason may be the lack of training among professionals working with infants and toddlers regarding the warning signs for early social-emotional difficulties [17]. Additionally, the availability of information for parents about social-emotional competencies and difficulties in young children is limited. The challenging nature of recognising internalizing type difficulties limits the amount of support services provided, putting young children at risk for continued and more severe difficulties in later childhood and into adolescence [5].

As development of social-emotional competence in infants occurs primarily in the family context in relationship with the primary caregiver(s) [18], it is necessary to explore the potential family and contextual correlates of early social-emotional adjustment. One variable which has been found to be closely correlated with social-emotional outcomes in young children is parenting self-efficacy. Parenting self-efficacy beliefs have emerged as a powerful predictor of competent parenting practices [19] and child social-emotional adjustment [20].However, the causal mechanisms involved in the relationship between parenting self-efficacy and child social-emotional adjustment are unclear. A reciprocal relationship likely exists between parents and children, with children impacting on parent efficacy estimations and parenting competence in a feedback loop [20]. Factors such as infant temperament and child behavioural adjustment have been directly linked to parenting competence [21,22]. Parenting self-efficacy has also been proposed to be an important mediator of the impact of various psychosocial factors on parenting competence and child adjustment. This highlights the importance of parenting self-efficacy as a potential buffer against the risks associated with various psychosocial factors such as socio-demographic status, social support, infant difficulty and maternal depression [19]. 

Parenting behaviour represents another salient factor in the development of social-emotional problems in young children. Parenting behaviour derives from a complex interplay between the parent and child within the social context [23,24,25,26].Current research supports the contention that both coercive parenting practices (harsh and inconsistent discipline) and low levels of warmth and responsiveness are predictive of the development of child behavioural difficulties [27,28].However, the direction of causation is as yet unknown, as several researchers have demonstrated that difficult child behaviour may also elicit poor parenting practices [27,29]. Optimal parenting has been conceptualized as responsive, sensitive or nurturant parenting. Responsive parenting is believed to play a critical role in children’s social-emotional development [30]. 

Maternal psychological distress may also contribute to the development of social-emotional problems and interrupt the acquisition of social-emotional competencies in young children. Much of the research to date has focused on parents with affective disorders, particularly depression [31]. More recent studies have explored the effects of different forms of maternal psychological distress and their timing [32]. These findings suggest that prenatal distress can have an adverse effect on cognitive, behavioral, and psychomotor development, and that postpartum distress contributes to cognitive and socio-emotional development [33]. Research has consistently demonstrated the negative impact of maternal psychological distress on parenting competence and child adjustment [33,34,35]. The impact of psychological distress on parenting behaviour is long term and enduring [36]. Parents who experience higher levels of emotional distress tend to display lower levels of responsiveness than non-distressed parents [31] and may be inadequate social partners for their infants [37]. The sustained effortful behaviour that parenting involves is also likely to prove more difficult for depressed parents, especially when their children are young [36]. Little is known about the processes that underlie the connection between parental emotional distress and competent parenting. However, a complex reciprocal relationship likely exists between mother and child with each contributing to each others current difficulties and risk for future difficulties [38]. 

An explicit aim of this study is to bring scholarly attention to infant and toddler development in an Irish context. It has been argued that bringing cultural concepts into conversation with psychological theories is, though a disputed process, necessary for rigorous scientific inquiry [39]. Specifically because development is a product of a child’s interaction with complex biological, cultural and social systems, it is imperative to examine development across various contexts [40]. While the last two decades have seen an increase in international research on infant social-emotional adjustment [41,42,43], there has been a comparative dearth of research in an Irish context. Though similar studies in American [44] and British [45] contexts have been conducted, no similar study has been conducted with Irish mothers and their young children. Especially considering the rapid social and economic changes that have occurred in Ireland, there is a need for an examination of social-emotional development in modern Irish infants and toddlers [46]. 

The current study aims to address the gaps in the research literature to date by;

Exploring rates of parent-reported social-emotional difficulties in young children in a clinical sample (referred to child services and awaiting clinical assessment) and infants in a non- clinical sample (attending mother and toddler groups in the community)Exploring predictors of social-emotional adjustment in young children within an Irish contextBuilding a predictive model of the relationship between parenting self-efficacy, parenting behaviour, maternal psychological wellbeing and social-emotional outcomes in infants and toddlers.

Based on the reviewed literature, we formulated two hypotheses. First we predicted that young childrenin the clinical sample would have higher reported levels of social-emotional problems and deficits in competencies than young children in the non-clinical sample., We hypothesized that mothers in the clinical sample would report higher rates of depression, anxiety and stress than mothers in the non-clinical sample. Understanding predictors of social-emotional difficulties in infants and toddlers which are amenable to change through intervention, may help to inform the development of effective interventions for young children experiencing and those at risk of developing social-emotional difficulties. 

## 2. Method

### 2.1. Participants

The participants in this study were mothers of young children aged between 12 and 48 months. The children in the clinical sample were on the waiting list for community psychology services or Early Intervention services across eight clinical catchment areas in the south-west of Ireland. This indicated that all of these children were presenting with difficulties which warranted referral for clinical assessment (and possibly intervention). In relation to the referral issues cited by mothers; the most common referral concerns were; speech and language delay (41%), regulatory difficulties (18%), behavioural difficulties (15%), and multiple concerns (13%). Participants in the comparison sample were young children of mothers who were attending community mother-toddler groups in the same catchment areas. These children were not on the waiting list for any child services. All services (clinical and community) were located across both urban and rural areas to reflect existing socio-economic diversity. 

One hundred and three eligible participants were identified and invited to take part in the clinical sample. Thirty-nine completed questionnaires were returned representing a response rate of 40%. Of the 110 eligible participants attending community mother-toddler groups, 33 completed questionnaires, were returned representing a 30% response rate in the non-clinical sample. 

An index of socio-demographic risk was calculated for the sample according to guidelines outlined by Sanders et al. [47]. 1. Mothers scored one point for reporting; (i) age of fewer than 20 years at the birth of target child, (ii) highest educational attainment of less than Inter (Junior) Certificate and (iii) gross family income of less than 20,000 euro annually. Following the scoring guidelines of Feehan et al. [48], 10% of mothers in the sample reported some risk (score of 1), while 4% reported significant socio-demographic risk (score of two or more), according to this index.

### 2.2. Design and Procedure

This study used a cross-sectional design to compare maternal reported social-emotional difficulties in young children in clinical and non-clinical samples and probe a predictive model. Ethical approval was granted by National University of Ireland, Galway and the Health Service Executive. Sealed envelopes containing the invitation letter, participant information sheet, consent form, questionnaire battery and a stamped addressed envelope were provided to the mother of each infant. Following data analysis feedback about the research findings was provided to participating mothers where requested and to participating services. 

### 2.3. Measures

Data in this study were collated using a questionnaire posted out to eligible mothers. The questionnaire comprised the following specific measures.

#### 2.3.1. Demographic Information

This section was compiled by the researcher and completed by the mother of each infant. Mothers were asked to provide: (1) socio-demographic information relating to the family, (2) information on child difficulties which led to a referral to clinical services (where relevant), (3) prior contact with child health services, (4) maternal mental health and (5) social support.

#### 2.3.2. Social-Emotional Adjustment

The Brief Infant-Toddler Social and Emotional Assessment (BITSEA: [49]) was used to assess the two criterion variables of interest in this study (social-emotional problems and social-emotional competencies). The BITSEA is a standardised assessment instrument used to identify children who may have social-emotional and behavioural problems and/or delays in the acquisition of social-emotional competence. BITSEA’s 42 items address social-emotional and behavioural problems in internalizing (‘Seems nervous, tense or fearful”) externalizing (“Is destructive. Breaks or ruins things on purpose”) and regulatory domains (“Wakes up at night and needs help to fall asleep again”). Items also assess maladaptive and atypical behaviours which may be indicative of other significant psychopathology (“Does not make eye contact”, “Hurts self on purpose”). BITSEA competence items assess the domains of attention, compliance, mastery, motivation, pro-social peer relations, empathy, imitation/play skills and social relatedness. Items include; “Shows pleasure when he or she succeeds” and “Plays well with other children (not including brother/sister)”. Each item is rated on a Likert scale from 0 to 2, where 0 = not true or rarely true, 1 = somewhat true and 2 = very true or often true. Scores are then summed to provide Problem and Competence Total scores. Cut-score values indicating Possible Problem correspond to approximately 25th percentile ranking or less. Cut-score values indicating Possible Deficit/Delay in Competencies correspond approximately to 15% percentile ranking or less. While high Problem scores or low Competence scores indicate the possible presence of problems and delays/deficits, follow-up assessment is required to determine whether these difficulties are clinically significant. The Parent Form also includes two questions that address; (1) the parent’s level of concern about the child’s behavior, emotions, or relationships and (2) the parent’s level of concern about the child’s language development. Responses to these questions do not count towards the BITSEA scores. Cronbach’s alpha for the BITSEA was 0.94. There is strong support for the construct validity of the BITSEA [49]. Moderate associations have been demonstrated between the BITSEA and other measures of social-emotional functioning, adaptive behaviour and developmental functioning [49].

The following questionnaires were used to explore possible predictors of social-emotional adjustment in infants; parenting behaviour, parenting self-efficacy (domain-general and task-specific) and maternal psychological distress. 

#### 2.3.3. Parenting Behaviour

The Parent Behaviour Checklist (PBC: [50]) was used to assess parenting behaviour in this sample. The PBC identifies parenting strengths and needs. PBC items assess parenting across three domains; Parental Expectations, Discipline and Nurturance. The Parental Expectations subscale assesses a parent’s developmental expectations (“My child should be able to use the toilet without help”). The Discipline subscale measures how a parent responds to difficult child behaviours (“I send my child to bed as a punishment”). The Nurturance subscale assesses strategies parents use to promote their child’s psychological growth (“I read to my child at least once a week”). The PBC provides three subscale scores within each of the domains rather than a total score. Interpretation of the PBC involves consideration of the validity of parent responses; namely do the responses accurately reflect parent’s actual expectations, nurturance and behaviour towards their young child [50]. When interpreting the PBC, consideration must be given to social desirability effects. Cronbach’s alpha for Discipline, Nurturance and Expectations were; 0.77, 0.9, and 0.87, respectively. 

#### 2.3.4. Satisfaction and Efficacy in Parenting Role

The Parenting Sense of Competence Scale (PSOC: [51]) is a 16-item self-report questionnaire designed to measure self-esteem in the parenting role. Statements about subjective perceptions of parenting are rated on a six-point Likert scale from 1 (Strongly agree) to 6 (Strongly disagree). Research has suggested that the PSOC is comprised of two distinct subscales representing parenting satisfaction and parenting efficacy (domain general) [51,52]. The satisfaction subscale examines parenting frustration, anxiety and motivation. Items include; “Being a parent makes me tense and anxious”. The efficacy subscale examines capability, problem solving ability and competence. Items include; “I honestly believe I have all the skills necessary to be a good parent to my child”. Each subscale was analysed separately for this analysis. For the purposes of this study, the focus was on the parenting satisfaction subscale. Cronbach’s alpha for parenting satisfaction was 0.82. The PSOC has been shown to have good construct validity.

#### 2.3.5. Confidence in Infant Care

The Toddler Care Questionnaire (TCQ: [53]) is a measure of maternal confidence or efficacy for the developmental issues which arise in infants. According to the TCQ, maternal confidence is defined as a mother’s perception of her effectiveness in managing a series of tasks or situations related to raising her toddler [54]. The TCQ is a task-specific measure of parenting efficacy; that is it specifically examines efficacy related to the particular tasks involved in parenting a toddler such as “knowing how to make your child feel better when he/she is upset” and “knowing when your child seems to want affection from you, such as a hug or a kiss”. The TCQ is comprised of 36 descriptive items such as these. The mother rates her feelings of efficacy or confidence for each on a Likert scale from 0 to 5 ranging from ‘very little confidence’ to ‘quite a lot of confidence’. The TCQ provides an overall score of maternal confidence/efficacy in the parenting role with higher scores indicating higher efficacy. Cronbach’s alpha for the TCQ was 0.94. There is evidence for the validity of the TCQ [54]. The TCQ was found to correlate with maternal depression and level of education completed [55].

#### 2.3.6. Negative Emotional Symptomatology 

In this study, the short form of the Depression, Anxiety and Stress Scale (DASS-21: [56]) was used to assess negative emotional symptomatology in mothers. The DASS 21 is a self-report measure used to identify and distinguish between subjective reports of depression (“I couldn’t seem to experience any pleasure at all”), anxiety (“I felt I was close to panic”) and stress (“I found it hard to wind down”). Individuals rate their level of distress over the past week on a 4 point Likert scale ranging from 0 (Did not apply to me at all) to 3 (Applied to me very much or most of the time). Items load onto three subscales; depression, anxiety and stress. The items within each scale are then summed to provide factor scores for Depression, Anxiety and Stress with higher scores indicating more severe depression, anxiety and stress. Cronbach’s alphas were 0.85 for Depression, 0.77 for Anxiety and 0.82 for Stress. There is evidence of good concurrent and discriminant validity for the DASS [57].

#### 2.3.7. Power Analysis

A priori power analysis was conducted using G*Power 3.1 [58] to estimate the required sample size for this study. The power figure was set at 0.80, with a medium effect size (0.15) and 8 predictor variables entered. This effect size was based on a review of previous research on parenting self-efficacy and social-emotional adjustment [20]. The recommended sample size for a study based on these criteria was 109.

## 3. Results

### 3.1. Overview

Data were normally distributed in clinical and non-clinical samples on the Parenting Sense of Competence scale (PSOC) only. Data were not normally distributed on the other predictor measures and the criterion measures in clinical and non-clinical samples. This reflects the general finding that clinical measures are often positively skewed in normal populations with people reporting few symptoms.

On the DASS 20% of mothers in the clinical sample and 15% in the non-clinical sample reported depression scores which were significantly higher than the normal range and may be clinically significant. In relation to anxiety, 26% in the clinical sample and 15% in the non-clinical sample reported scores which may be clinically significant. On the Stress subscale, 26% in the clinical sample and 15% in the non-clinical sample reported stress scores which may be clinically significant.

Concerning the criterion measure of social-emotional problems, 55.5% of young children in the clinical sample and 15% in the non-clinical sample scored above the cut-score indicating possible clinically significant social-emotional difficulties. Of these mothers in the non-clinical sample, 33.3% indicated ‘no concern’ regarding their children.

Concerning the second criterion variable; social-emotional competencies, 55.5% of young children in the clinical sample and 30% in the non-clinical sample scored below the cut score indicating possible clinically significant deficits in social-emotional competencies. Differences between the clinical and non-clinical groups on the predictor and criterion variables were explored using t-tests for normally distributed data and Mann Whitney *U* tests for non-normally distributed data. The results are presented in Table 1. Concerning the predictor variables, significant differences were found between mothers in the clinical and non-clinical samples on measures of parenting satisfaction [*t*(70) = −1.69, *p* < 0.05], parenting nurturance *(U* = 475.5, *z* = −1.91, *p* < *0*.05), task-specific self-efficacy (*U* = 459, z = −2.09, *p* < *0*.05) and anxiety *U* = 506.5, *z* = −1.65, *p* ≤ *0*.05). The direction of difference was as expected with mothers in the non-clinical group scoring higher on parenting satisfaction, parenting nurturance, task-specific self-efficacy and lower on anxiety. No significant differences were found between participants in the clinical and non-clinical samples on other predictor measures. There were also highly statistically significant differences between the clinical and non-clinical sample on the criterion measures in the expected direction. Young children in the clinical sample had higher reported social-emotional problems (*U* = 258.5, *z* = −4.04, *p* < *0*.01) and lower competencies (*U* = 381, *z* = −2.57, *p* < *0*.01) than those in the non-clinical sample. 

### 3.2. Correlation Matrix

Spearman Rank correlations were conducted to determine the inter-correlation of the predictor and criterion variables. Several statistically significant correlations were found which justified further exploration using regression analysis. Parenting satisfaction (*rs* = −0.461, *p <* 0.01), maternal depression (*rs* = 0.469, *p <* 0.01), anxiety (*rs* = 0.543, *p <* 0.01), and stress (*rs* = 0.457, *p <* 0.01) were all positively correlated with infant social-emotional problems. Task-specific self-efficacy (TCQ) was negatively correlated with infant social-emotional problems (*rs* = −0.258, *p* < 0.05).

A number of significant relationships were also observed between the predictor variables and the second criterion variable; social-emotional competencies. Parenting nurturance (PBC) (*rs* = 0.216, *p <* 0.05), parenting expectations (*rs* = 0.208, *p <* 0.05) and task-specific self-efficacy were positively correlated with social-emotional competencies in infants (*rs* = 0.242, *p <* 0.05). 

### 3.3. Hierarchical Multiple Regression

Hierarchical multiple regression was then used to test the predictive validity of two proposed models with social-emotional problems and social-emotional competencies as the respective criterion variables. Initially, all the predictor variables which correlated significantly with the criterion variable were entered into the model in blocks, according to their theoretical importance. The grouping variable (clinical or non-clinical group) was entered in the first block. Parenting variables were entered in the second block. Indicators of maternal psychological wellbeing were entered in the third block. The results of the analyses suggested a statistically significant model fit between the predictor and criterion variables and indicated the amount of variance accounted for by the models.

### 3.4. Predictive Model 1: Social-Emotional Problems

The first multiple regression was carried out with social-emotional problems as the criterion variable.

The results of this analysis are shown in Table 2. The results suggested a statistically significant model fit between the three blocks of predictors and the criterion variable (*R*^2^ = 0.59, *F* (8, 60) = 10.49, *p* < 0.001). The model accounted for 59% of the variance in social-emotional problems in infants. Clinical or non-clinical group status (*β* = −0.34, *t* = −3.86, *p* < 0.001), parenting satisfaction (*β* = −0.25, *t* = −2.03, *p* < 0.05), maternal depression (*β* = 0.40, *t* = 2.83, *p* < 0.01), and maternal anxiety (*β* = 0.29, *t* = 2.17, *p* < 0.05) all significantly predicted social-emotional problems in young children. Parenting behaviour (discipline and nurturance), task-specific self-efficacy and maternal stress were not significant predictors of social-emotional problems. The magnitude of the standardised beta weights suggests that maternal depression made the greatest contribution to the model. There was a positive correlation between maternal depressive symptoms and social-emotional problems in infants and toddlers. A similar relationship was observed concerning maternal anxiety, suggesting that increased anxiety was associated with more reported child social-emotional problems. Additionally, as parenting satisfaction increased, child social-emotional problems decreased.

### 3.5. Predictive model 2: Social-Emotional Competencies

A second multiple regression was carried out with social-emotional competencies as the criterion variable. The results are shown in Table 3.

The results suggested a statistically significant model fit between the predictors and the criterion variable (*R*^2^ = 0.21, *F* (1, 64) = 5.09, *p* < 0.05). The model accounted for 21% of the variance in social-emotional competencies in infants. Clinical or non-clinical group status was statistically significant in the first step of the model (*β* = 0.30, *t* = 2.59, *p* < 0.05) and contributed 9% to the power of the model in predicting social-emotional competencies. However, by the final stage of the model, clinical or non-clinical group membership no longer made a significant contribution to the model. Task-specific self-efficacy (*β* = 0.28, *t* = 2.26, *p* < 0.05) was the only significant predictor of social-emotional competencies in young children in the final model. Parenting expectations and parenting nurturance were not significant predictors of child social-emotional competencies. The direction of the standardised beta coefficients suggested that as task-specific self-efficacy increased, child social-emotional competencies also increased. 

## 4. Discussion

The current study aimed to address the gap in the research literature to date by exploring rates of social-emotional adjustment difficulties in young children and associated maternal variables within an Irish context. There is growing recognition of the importance of assessing social-emotional adjustment in infancy and an increasing discussion of the necessity to consider sociocultural developmental contexts. According to the research literature, early deficits in social-emotional functioning are relatively stable [1,2] and represent a significant risk factor for depressive symptoms, oppositional defiant or conduct disorder, poor peer relationships, social skills deficits, poor academic performance and psychiatric problems later in life [59]. 

The first objective of the current study was to compare rates of social-emotional difficulties in young children in a clinical and non-clinical sample. In the current study, 55.5% of young children in the clinical sample had significant levels of social-emotional problems and/or significant delays in social-emotional competence. Similarly, in a study of Early Intervention services in the U.S., Briggs-Gowan and Carter [60] reported that up to 60% of children had high problems and/or low competence in social-emotional domains. In the current study, 15% of young children in the community sample had significant social-emotional problems and 30% had significant delays in the acquisition of competencies. Briggs-Gowan and Carter [60] reported rates of 31% of infants with high problems and/or low competencies in their normative sample. The consistency in findings between the current study and previous research lends support to the reliability of the findings. However, there is limited international data available on the prevalence of social-emotional difficulties in infants, particularly in non-clinical samples. 

Significant social-emotional problems were reported for infants and toddlers in both clinical and non-clinical samples in the current study. One third (33.3%) of mothers of young children in the non-clinical sample who scored above the cut-score for social-emotional problems indicated ‘no concern’ regarding their infants. This suggests that parents may have difficulty in recognizing the problems they endorsed, as possible indicators of risk. Parents in the current sample may be interpreting the social-emotional difficulties of their young children as part of the normative developmental process. Failure to accurately recognize such difficulties as possibly clinically significant may lead to a continuation and increase in the severity of symptoms and delay access to clinical assessment and treatment. This can have a long-term negative impact on child health and wellbeing. This may also impact on parent functioning contributing to psychological distress, poor parenting satisfaction and decreased feelings of efficacy in the parenting role as parents feel they have little control over child outcomes. 

The second objective of this study was to identify predictors of social-emotional outcomes in young children and build a predictive model of child social-emotional adjustment. The results of a hierarchical multiple regression with social-emotional problems as the outcome variable demonstrated that clinical/non-clinical group membership, parenting satisfaction, maternal depression, and maternal anxiety were significant predictors of child social-emotional problems in a model which explained 59% of the variance. Parenting self-efficacy, parenting behaviour and maternal stress were found to be non-significant predictors. Maternal psychological distress (depression and anxiety) was the most significant contributor to the model of child social-emotional problems, adding 15% to the variance explained. This aligns with previous research, which has demonstrated the importance of maternal psychological distress in predicting children’s behavioural and social-emotional adjustment [34,35]. However, little is known about the mechanisms through which maternal psychological distress impacts on child adjustment.. The impact of maternal depression may be greatest for infants who have the smallest repertoires of behaviour and are the most dependent on parents for their care [36]. This may explain the highly significant contribution of maternal psychological distress in predicting child social-emotional problems demonstrated in the current model. While maternal psychological distress may contribute to the development of social-emotional problems in young children, it is also likely that children with social-emotional difficulties place greater demands on parent resources, which may contribute to the development of psychopathology. 

In the current study 20% of mothers in the clinical sample and 15% in the non-clinical sample reported depression scores which may represent clinically significant depression. In relation to anxiety, 26% in the clinical sample and 15% in the non-clinical sample reported scores for anxiety which are above the normal range and may be clinically significant. Both the high rates of maternal psychological distress reported in the current study and the significance of this variable in predicting child social-emotional problems have important implications for intervention. This emphasizes the need for assessment and intervention with young children to be comprehensive and family focused. These findings suggest the importance of concurrently assessing and monitoring the psychological wellbeing of mothers and young children presenting to services.

Parenting satisfaction was also found to be a significant predictor of social-emotional problems in young children. Previous research has supported the link between lower parenting satisfaction and child problem behaviour [51,52,61]. A complex reciprocal relationship likely exists between mother and child with each contributing to each other’s current adjustment and risk for future maladjustment [37]. This finding has important clinical implications for children experiencing social-emotional problems and for the services providing support to these children and their families.

A second model of child social-emotional competencies found that parenting self-efficacy was the only significant predictor. Overall, the model explained 21% of the observed variance in social-emotional competencies in young Irish children. The potency of the self-efficacy construct, measured at a task-specific level, in predicting parenting competence and social-emotional adjustment in infants, has been supported in previous research [19]. However, differences in the operationalization of the construct of social-emotional adjustment make comparisons across studies more difficult. Coleman and Karraker [19] found that task-specific parenting self-efficacy was significantly related to positive child social-emotional adjustment in a sample of predominantly middle-class, mother–toddler dyads. In their study, an average of 12% of the variance in child outcomes was explained by variability in parenting self-efficacy beliefs [62]. In the current study, a stronger predictive model was developed for social-emotional problems than competencies in young children. Perhaps this reflects the focus of the current study on a deficit-based model. The psychometric measures utilized may be more conceptually linked to social-emotional problems than competencies.

In the current study, parenting behaviour (discipline and nurturance) was not found to be a significant predictor of social-emotional problems or competencies in young children. Previous research regarding the predictive power of parenting behaviour concerning child social-emotional adjustment has been mixed. Gutermuth Anthony et al. [63] explored parenting behaviour as a predictor of social-emotional adjustment in preschoolers in a large clinical and non-clinical sample in the U.S. Results did not support a link between parenting behaviour and child maladjustment [63]. Concerning parenting nurturance specifically, a body of research has provided support for the role of responsive and nurturant parenting in promoting infant social-emotional adjustment [30,64]. One possible suggestion for the lack of findings regarding the role of nurturant parenting in social-emotional adjustment in young children in the current study may be the methodological limitations of using a questionnaire-based measure. Previous research has also demonstrated a relationship between parental use of aversive disciplinary strategies and child maladjustment [65]. However, while research supports the role of parental discipline style in predicting child externalizing behaviour, the relationship between parental discipline and internalizing difficulties is less well known. This may explain why aversive discipline was not found to be a significant predictor of child social-emotional adjustment in the current study. It is also worth noting that very low levels of aversive discipline were reported by mothers in the current sample. This is in contrast to previous studies which have suggested high levels of aversive discipline particularly in mothers of children in clinical samples [65]. This may reflect the operation of positive response bias.

There are a number of limitations in the current study. One is the small sample size. A total of seventy-two mother-child dyads across clinical and community samples participated in the research. The limited sample size may reflect the difficulty of engaging mothers of young infants and toddlers in research due to the high parenting demands during this developmental period. Another limitation of the current research concerns the issue of shared method variance. All of the predictor variables and the outcome variables in this study were determined based on maternal self-report. This introduces the possibility of response bias. Previous research has indicated a possible impact of maternal psychological distress on the reporting of child adjustment difficulties [31]. Mothers who are experiencing concurrent psychological distress are more likely to find child behaviour problematic which may lead to an overestimation of problem scores. Given the high rates of maternal psychological distress reported for the current sample, there may be a possibility of over-inflated reports of child adjustment difficulties. In the current study, as expected, many of the clinical measures were positively skewed in the non-clinical sample. Although robust to deviations from normality, the use of parametric regression analysis may have resulted in inflated confidence intervals, thus reducing the likelihood of detecting statistically significant differences [66,67]. Future studies with greater statistical power, may detect an even stronger relationship between independent and dependent variables than reported here [68,69]. 

## 5. Conclusions

To date, no other studies have explored the prevalence of social-emotional difficulties in infants and toddlers in an Irish context. The current study addresses this significant gap in the research literature and provides a baseline from which to monitor and compare prevalence rates. Further research in the Irish context is needed to determine whether the rate of social-emotional adjustment difficulties found in the current sample is reflective of the broader national picture. While the high rates of social-emotional adjustment difficulties in young children are concerning due to the negative impact on child health and wellbeing, it also corresponds with the available international data. This study also enhanced the limited body of knowledge regarding predictors of social-emotional adjustment in young children less than four years of age. Several significant predictors emerged; parenting satisfaction, task-specific self-efficacy, maternal depression, and maternal anxiety. It is important to note that each of these predictors is amenable to change through intervention. In this way, these findings have pragmatic value for clinicians working with young children and families. 

Further research is needed to cross-validate the proposed models of social-emotional problems and competencies in young children found in the current study, which would facilitate broader generalisation of the findings. Future research should target some of the limitations of the current study, particularly concerning sample size, self-report measures, and statistical power. Consideration should also be given as to how to successfully encourage mothers of young children to participate in research. This could contribute to informing clinical practice to better meet the needs of mothers and young children at risk for social-emotional difficulties. 

## Figures and Tables

**Table 1 ijerph-18-02861-t001:** Overview of Questionnaire measures.

Measure	Clinical Sample (*n* = 39)		Non-Clinical Sample (*n* = 33)			*p*
	Mean	SD	Mean	SD	*t*(df)/*U*	
PSOC Satisfaction	37.44	7.51	40.83	6.94	−1.69(70)	0.048
DASS Depression	2.90	3.26	2.27	3.12		
DASS Anxiety	2.10	2.67	1.20	2.19	506.5	0.05
DASS Stress	5.28	3.26	4.30	3.21		
PBC Expectations	25.74	6.69	27.94	8.71		
PBC Discipline	13.97	5.26	13.0	5.22		
PBC Nurturance	31.92	5.26	34.12	4.13	475.5	0.028
TCQ Parenting Efficacy	149.21	21.40	160.03	13.94	459.0	0.018
BITSEA Problems	15.67	9.91	8.24	6.28	258.5	0.00
BITSEA Competencies	15.06	3.70	17.15	2.94	381.0	0.005

**Table 2 ijerph-18-02861-t002:** Multiple Regression Statistics: Model 1.

Predictor	B	Std. Error	Beta	*p*<	*R* ^2^
Block 1					
Constant	22.974	3.118			
Non/Clinical sample	−7.445	2.023	−0.410	0.001	0.17
Block 2					
Constant	35.310	9.883			
Non/Clinical sample	−6.374	1.835	−0.351		
PBC Nurturance	0.031	0.236	0.017		
PBC Discipline	−0.016	0.219	−0.008		
PSOC Satisfaction	−0.695	0.137	−0.561		
TCQ Total	0.079	0.054	0.166	0.001	0.43
Block 3					
Constant	18.857	10.005			
Non/Clinical sample	−6.172	1.600	−0.340	0.001	
PBC Nurturance	0.031	0.206	0.017		
PBC Discipline	0.049	0.192	0.025		
PSOC Satisfaction	−0.307	0.152	−0.248	0.05	
TCQ Total	0.064	0.047	0.134		
DASS Depression	1.128	0.399	0.395	0.01	
DASS Anxiety	1.061	0.490	0.289	0.05	
DASS Stress	−0.462	0.437	−0.165	0.001	0.59

**Table 3 ijerph-18-02861-t003:** Multiple Regression Statistics: Model 2.

Predictor	B	Std. Error	Beta	*p*<	*R* ^2^
Block 1					
Constant	12.993	1.250			
Non/Clinical sample	2.102	0.811	0.302	0.05	0.091
Block 2					
Constant	2.344	3.845			
Non/Clinical sample	1.243	0.825	0.179		
TCQ Total	0.051	0.022	0.276	0.05	
PBC Expectation	0.091	0.051	0.201		
PBC Nurturance	0.050	0.087	0.070	0.05	0.211

## Data Availability

10.6084/m9.figshare.14186105.

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
