# Peer review of "Building a Predictive Model of Social-Emotional Adjustment: Exploring the Relationship between Parenting Self-Efficacy, Parenting Behaviour and Psychological Distress in Mothers of Young Children in Ireland"

_ijerph, 2021, doi:10.3390/ijerph18062861_

Round 1

Reviewer 1 Report

Dear authors,

Thank you for the interesting manuscript. I had good time reviewing it!

The study examines the relationship between several maternal characteristics (parenting behavior, parenting self-efficacy and parenting stress) and infant social-emotional problems. I think the manuscript is in general well-written.  However, I have few suggestions and comments mostly in regard of statistical analysis and interpretations.

  1. The term “hierarchical regression” is actually served for a hierarchical nested data: children nested in the class or school. I believe the term step-wise regression is more appropriate for the analysis the authors have performed.
  2. Line 45: ten%, better: 10%. Line 51: please put “.” at the end of the sentence. Please re-check the manuscript thoroughly in this regard.
  3. Please provide examples of items for each measure. Having the items examples may help readers to understand the operationalization of the measures. Sometimes they are also useful to understand the discussion part.
  4. Line 340-342: one third (33.3%) of mothers of infants with possible clinically significant social-emotional problems indicated ‘no concern’ regarding their infants. This suggests that parents may have difficulty in recognizing these symptoms as indicators of risk. What are the “symptoms” listed in the questionnaire? Since the instrument cannot be used as a clinical diagnostic, one should not directly label the “non-concern” parents as “having difficulty in recognizing the symptoms”. In addition, the “non-concern” construct was not explained in the manuscript (and to my understanding is not part of study’s focus). Therefore, it would be better to include the measure and finding in the respective parts of the manuscript if authors see the necessity of explaining these results in the discussion part.
  5. A consistent approach towards interpretation of multiple regression should be implied. In line 2 and 30 authors mentioned about predictive relevance of various parents’ variables on child’s socio-emotional development. However, in discussion (e.g., line 366 to 370 and 383-384), authors change their perspectives and refer the results of multiple regression as correlational in nature. A significant modification is needed in order to deliver clear messages for the readers.

Author Response

Dear reviewer,

Thank you kindly for taking the time to read and review my article. I really value your comments and suggestions and hope I have addressed any concerns to your satisfaction.

  1. We used HRM rather than Stepwise Regression as our study is theoretically informed. HRM allowed us to enter variables into the equation in an order specified by us. Stepwise Regression would have forced the variables into the equation based on statistical criteria only.

  2. Line 44: Ten% has been amended to 10% here and throughout the article.

Line 47: “.” Has been added at the end of the sentence and throughout the article.

  1. I have provided examples of items from each measure. See Line 177-184 as an example.
  2. This has been clarified within the description of the BITSEA measure in the Method and subsequently in the Results and Discussion sections. Line 191-194, Line 273-274 and Line 376-382.
  3. This research focussed on developing a predictive model for infant social-emotional adjustment. The results have been edited to focus on discussion of the observed findings and the contribution of each of the variables in predicting social-emotional adjustment, within the context of previous research (line 397-404 and line 414-419) .

Reviewer 2 Report

I thank the editor for the possibility to revise the paper titled “Building a predictive model of infant social-emotional adjustment: Exploring the relationship between parenting self-efficacy, parenting behaviour and psychological distress in mothers of infants in Ireland.” 

The aim of the study was to define a predictive model about the children’s social-emotion adjustment which would be applied in prevention and intervention setting. 

The study constitutes a contribution in the developmental psychology field. However, I believe it needs major improvements before its publication. My main concerns revolves around the statistical analysis, which were inadequate for the purpose of this study: 

Please find below my suggestions. 

Introduction section:

  1. Overall, the references in the introduction paragraph should be uploaded consistently. Also, when the authors stated “international literature” (line 42) they did not report them. 
  2. Line 50: the authors could better explain what “developmental outcomes” were predicted by deficit in social-emotional functioning. 
  3. Line 93: Again, I suggest the authors to refer to the recent evidence about the negative impact of the maternal distress on child’s adjustment. For example, I suggested these recent articles that highlight this issue: 
  4. Hossain MM, Sultana A, Purohit N. Mental health outcomes of quarantine and isolation for infection prevention: a systematic umbrella review of the global evidence. Epidemiol Health.(2020) 42:e2020038. doi: 10.4178/epih.e2020038
  5. Lecciso F, Levante A, Antonioli G, Petrocchi S. The effect of dyadic trust and parental stress on the child's resilience: a comparison between heterosexual and homosexual families. Life Span Disabil.(2020) XXIII:85–108.
  6. Petrocchi, S., Levante, A., Bianco, F., Castelli, I., & Lecciso, F. (2020). Maternal Distress/Coping and Children's Adaptive Behaviors During the COVID-19 Lockdown: Mediation Through Children's Emotional Experience.
  7. Wang C, Ng C, Brook R. Response to COVID-19 in Taiwan: big data analytics, new technology, and proactive testing. JAMA.(2020) 323:1341–2. doi: 10.1001/jama.2020.3151
  8. The study’s purposes, in the last section of the introduction paragraph, could be summarized in bullet point. This would make the section more readable. Furthermore, reading the study’s purpose I did not immediately understand whether or not the “infant in the clinical sample” means that children were affected by psychopathology. Later, the authors specified that the clinical sample included mothers affected by pathology, but I think, it would be useful to clarify this issue before.  

Method section:

The section should be improved consistently. The Authors did not report the sample descriptive info: i.d., mean and standard deviation of mothers and children’ age; were the children preterm or term? What was the mothers’ educational level? What pathology affects mothers? Etc etc. 

I think these are pivotal info describing the sample. Please summarize them. 

Measures were well described. Line 198, there was a typo. 

Results:

Usually the decimal values are reported without the 0 before the dot.

Line 249, there was a typo. 

Overview: in table 1 the t-values and the U-values were not reported. It’s important to know what variables were normally or non-normally distributed. 

For a better readability of the results, I suggest the authors to summarize the correlation results. 

Line 281, the authors could separate the paragraph regarding the correlation and the one regarding the hierarchical regression. 

Regarding the statistical analyses that the authors carried out to test the models, I have some concerns. Firstly, there was no graphic representation that would help to better “visualize” the model testing. Secondly, they did not test the model through structural equations. I believe that a model that should be used for prevention and intervention purposes should be tested using a robust statistical analyses. Thus, I suggest to re-computed the analysis or to re-think the title/purposes of the paper. Thirdly, the authors carried out parametric regression on a small sample: I think that non-parametric regression applying RStudio packages (e.g., psych, mbml, or others) could improve the results. 

The Discussion and limitation sections were clear and detailed. An uploaded references search and further statistical analyses could improve the paper. 

Author Response

Dear reviewer,

Thank you kindly for taking the time to read and review my article. I really value your comments and suggestions and hope I have addressed any concerns to your satisfaction.

  1. References have been uploaded consistently throughout the article and international literature has been reported Line 42 [4-15].
  2. Examples of developmental outcomes have been included. Line 46-47.
  3. I have included evidence from a number of recent articles about the negative impact of the maternal distress on children’s adjustment including a systematic review in this area; [41, 42]. See Line 92
  4. The study’s purpose has been summarized in bullet point (line 112-117). The issue regarding ‘clinical sample’ has now been clarified in the aims of the study’. Line 112-115. Descriptive information on background mental health difficulties of mothers has been included in the Method section (line133-135).
  5. Sample descriptive information has been included (Line 128-145).
  6. Line 212 typo has been corrected.
  7. The decimal values have been reported without the 0 before the dot in Results section.
  8. Line 272 This typo has been corrected.
  9.  I have included this information (t-values and U-values) in Table 1 (line 292-293 ).
  10.  The correlation results have been summarized more concisely (line 295-304).
  11.  The paragraph regarding the correlation and the one regarding the hierarchical regression have been seperated (line 305).
  12. i: Graphic representation of the model testing has been included (Line 319-321 and 340-342).

    ii: The purposes of the current paper have been revised with the stated aim being to generate greater understanding of social-emotional difficulties in infants in an Irish context.

    iii: In relation to this concern, it is acceptable to use multiple regression when the outcome variable is not normally distributed. Other assumptions concerning sample size, multicollinearity and distribution of scores were not violated, suggesting that the regression model used was appropriate for the present sample and applicable to the population (Field, 2005). Also see

    O'Sullivan, D. J., O'Sullivan, M. E., O'Connell, B. D., O'Reilly, K., & Sarma, K. M. (2018). Attributional style and depressive symptoms in a male prison sample. PLoS ONE, 13(2), Article e0190394.

    This has also been taken into consideration when drawing inferences from the findings.

Reviewer 3 Report

Basically I found the paper very relevant, with a very good introduction, making the point of the lack of studies based on Ireland - which I found very relevant -, literature references were also appropriated, so the research questions.

About method, sampling was very good, and I found only these minor errors, which they must change and checked it; data analysis was appropriate as well, so it was the discussion.

Thanks for give me the opportunity to review this paper, which I consider very appropriate to be published.

Only two issues

1.- there is a mistake in the line 144; authors indicate 3%, and it should be 30%

2.- in the line 151 authors indicate 4% and it sounds too low, this number should be check it.

and that is all.

Author Response

Dear reviewer,

Thank you kindly for taking the time to read and review my article. I really value your comments and suggestions and hope I have addressed any concerns to your satisfaction.

1.The mistake in the line 149 has been corrected; "30% response rate in the non-clinical sample".

2. This has been checked and numbers of mothers reporting ‘some’ socio-demographic risk have been included to provide more clarity. “Following Feehan et al. [54], 10% of mothers in the sample reported some risk (score of 1), while 4% reported significant socio-demographic risk (score of two or more), according to this index” (line 153-155).